



# The effect of soil moisture content and soil texture on fast in situ pH measurements with two types of robust ion-selective electrodes

Sebastian Vogel[1], Katja Emmerich[1], Ingmar Schröter[2], Eric Bönecke[3], Wolfgang Schwanghart[4], Jörg Rühlmann[3], Eckart Kramer[2], Robin Gebbers[1,5]

[1]Department of Agromechatronics, Leibniz Institute for Agricultural Engineering and Bioeconomy (ATB), Max-Eyth-Allee 100, 14469 Potsdam, Germany (ORCID Sebastian Vogel: 0000-0002-9625-8510)
[2]Eberswalde University for Sustainable Development, Landscape Management and Nature Conservation, Schicklerstraße 5, 16225 Eberswalde, Germany
[3]Leibniz Institute of Vegetable and Ornamental Crops (IGZ), Theodor-Echtermeyer-Weg 1, 14979 Grossbeeren, Germany
[4]Institute of Environmental Sciences and Geography, University of Potsdam, Potsdam, Germany
[5]Chair of Agricultural Business Operations, Martin-Luther University Halle-Wittenberg, Karl-Freiherr-von-Fritsch-Strasse 4, 06120 Halle, Germany

*Correspondence to*: Sebastian Vogel (svogel@atb-potsdam.de)

Keywords: Soil pH value, digital soil mapping, pH sensor, proximal soil sensing

**Abstract.** In situ soil pH measurements with ion-selective electrodes (ISE) receive increasing attention in soil mapping for precision agriculture, as they can avoid time consuming sampling and off-site laboratory work. However, unlike the standard laboratory protocol, in situ pH measurements are carried out at lower and varying soil moisture content (SMC), which can have a pronounced effect on the sensor readings. In addition, as the contact with the soil during in situ measurements should be relatively short, effects of soil texture could be expected, because texture controls the migration of protons to the electrode interface. This may be exacerbated by the fact that the electrodes used for in situ measurements are made of less sensitive but more robust materials as compared to the standard glass electrode. Therefore, the aim of the present study was to investigate the effect of soil moisture and soil texture on pH measurements using robust antimony and epoxy-body ISE pressed directly into the soil for 30 seconds. The SMC was gradually increased from dry conditions to field capacity. A wide range of soil texture classes were included with sand, silt and clay contents ranging from 16 to 91 %, 5 to 44 % and 4 to 65 % respectively. An exponential model was fitted to the data to quantify the relationship between SMC and pH. The results show that an increase in SMC causes a maximum increase in pH of approximately 1.5 pH units, regardless of the type of pH ISE used. Furthermore, for sandy soil textures, a rather linear relationship between pH and SMC was observed, whereas with decreasing mean particle diameter (MPD), the model had a pronounced exponential shape, i.e. a greater pH increase at low SMC and a plateau effect at high SMC. With increasing SMC, the pH values asymptotically approached the standard pH measured with a glass electrode in 0.01 M $CaCl_2$ (soil:solution ratio = 1:2.5). Thus, at high SMC, subsequent calibration of the sensor pH values to the standard pH value is negligible, which may be relevant for using the sensor pH data for lime requirement estimates. The pH measurement error decreases exponentially with increasing soil moisture and increases with decreasing MPD. Using a knee point detection, reliable pH values were obtained for SMC > 11%, irrespective of the pH ISE used. An analysis of the regression coefficients of the fitted exponential model showed that the maximum pH increase also depends on soil texture, i.e. the



influence of soil moisture variation on the pH value increases with decreasing MPD. Moreover, the concavity of the exponential curve increases with decreasing MPD.

# 1 Introduction

In agriculture, soil acidity is one of the fundamental soil properties for characterizing soil fertility and soil health because it directly and indirectly controls a series of soil physical, chemical and biological properties important for plant growth (Robson, 1989; Epstein and Bloom, 2001; Mengel and Kirkby, 2002). Since extreme acidic or alkaline conditions can constitute undesirable yield-limiting factors, an accurate assessment of soil acidity by measuring the pH value of arable soils is a prerequisite to sustain or increase crop productivity. In humid climates, where soils naturally tend to acidify, soil acidity can

be managed by means of lime fertilization. Precise soil acidity management should comply with site-specific lime demands, which requires pH data in a high spatial resolution (Brouder et al., 2005; Gebbers et al., 2009). However, conventional grid sampling and standard laboratory pH analyses are too tedious and expensive. As an alternative, mobile pH sensors are increasingly applied for fast and cost-effective in situ pH measurements (Adamchuk and Lund, 2008; Adamchuk et al., 1999; Viscarra Rossel et al., 2005; Viscarra Rossel and McBratney, 1997; Schirrmann et al, 2011; Bönecke et al., 2020). The pH

sensor system should enable frequent and fast measurements (e.g., less than 1 minute) in order to map a field within reasonable time. Moreover, the system must be robust, and the results must be comparable to the standard lab-based pH measurements. To meet these requirements, strategies should include the reduction of sample preparation and measurement time (Adamchuk et al., 1999; Adamchuk and Lund, 2008), as well as the use of different sensors as compared to the standard lab procedure, e.g., colorimetric approaches, ion-selective field effect transistors (ISFET) or metallic ion-selective electrodes (ISE)

(Adamchuk and Lund, 2008; Viscarra Rossel and McBratney, 1997; Yuqing et al., 2005).

The pH ISE with a bulb-type glass membrane is the standard sensor for measuring pH in the laboratory (Thomas, 1996; Essington, 2015). It is composed of a measuring electrode that responds to changes in the hydrogen ion (H+) concentration of a sample solution via a thin H+-sensitive membrane and a reference electrode providing a constant electrical potential. During measurement, a pH-dependent potential is generated between the sample solution and the membrane. The potential difference,

determined between the measuring and the reference electrode, is linearly related to the pH value of the sample according to the Nernst Equation (Thomas, 1996; Essington, 2015). A high positive potential implies a high H+ concentration and low pH while a low (negative) potential implies a low H+ concentration and a high pH value. The membrane of a standard pH ISE for laboratory use is made of glass, making them very fragile and unsuitable for in situ soil pH measurements. Thus, colorimetric approaches, ISFET, and metallic electrodes were investigated as possible alternatives (Viscarra Rossel and McBratney, 1997)

and glass electrodes were ruggedized. Among these, pH ISE based on antimony turned out to have favourable properties, even though some manufacturers also suggest glass electrodes with flat-surface tips and with epoxy bodies for measurements in difficult environments (like soil). Several studies have compared the performance of antimony and glass electrodes and showed



good agreement when measuring the pH value in soil solutions (e.g. Conkling and Blanchar, 1988; Baghdady and Sommer, 1990; Decker et al., 2017).

Besides the type of electrode, another fundamental disparity between the laboratory and in situ measurements of pH is the pre-treatment of the sample. In the lab, soil samples are standardized by drying and sieving to < 2 mm. In contrast, in the field, the measurement conditions are affected by spatially and temporally fluctuating soil moisture content (SMC), which is highly influenced by soil texture and topographic position. In addition to the mechanical stress and variable moisture conditions, in situ measurements can create problems due to losses of the inner aqueous electrolyte solution to the unsaturated soil, drying

of the glass membrane and the suspension effect (Thiele-Bruhn et al., 2015). The suspension effect is named after the observation that pH readings are different from clear solutions of soil extracts (filtrates or the clear supernatant solution of suspensions) as compared to measurements in the sediments of a soil suspension (Essington, 2015). According to Sumner (1994) and Essington (2015), the cation exchange capacity (CEC) of the soil effects the mobility of $K^+$ ions in the $K^+ - Cl^-$ salt bridge of the potentiometric electrode system. In soil suspensions with high CEC, $K^+$ ions can be attracted by the negatively

charged soil colloids and will move faster trough the salt bridge than the $Cl^-$ ions. This can result in a lower pH reading in the suspension as compared to the clear supernatant. Conversely, in suspensions with low CEC (due to high amounts of hydrous Fe and Al at low pH), the mobility of $Cl^-$ through the liquid junction may be greater than that of $K^+$ resulting in higher pH readings. While the suspension effect can be controlled in laboratory measurements this is not the case for in situ measurements.

Previous studies have investigated the effect of varying SMC on the pH measurement on different soils (Schaller and Fischer, 1981; Adamchuk et al., 1999; Kahlert et al., 2004; Oliviera et al., 2018; de Souza Silva and Molin, 2018; Patil et al., 2019). However, most of the studies have been carried out with glass electrodes and detailed evaluation of antimony electrode performance at low moisture regimes in different soil textures is lacking.

Early research on the SMC-pH relationship for glass electrodes was reported by Keaton (1938) and Davis (1943). Keaton

(1938) observed a pronounced influence of the soil-water ratio on the pH readings in his laboratory experiments. He found a pH decrease of up to 2 units with the reduction of SMC from 1000 % (1:10 ratio) to saturation while the pH rises again when the SMC was further reduced to field moisture. He explained this with the low enough proportion of metallic cations to H+ to produce counteraction of the Debye-Hückel activity effect by the effect of preferential dissociation. Davis (1943) investigated pH measurements in soils with SMC at and below 100 %. The pH values obtained immediately after inserting the electrode

into the soil increased with SMC and levelled out at about 35 %. However, readings for the electrode exposed to the soil for 14 hours produced an inverse relationship. The pH from dryer soils (below 35 % SMC) has jumped to very high values and the pH has dropped with increasing SMC. This, along with a lack of reproducibility in dryer soils, led Davis (1943) to the conclusion that pH should not be measured in soils below moisture equivalent. He explained the elevated inaccuracy of measurements partly with lacking soil contact and problems of the amplifier in the measuring instrument to handle the high

resistance. Thus, he concluded that there is no evidence that dry soils are characteristically more acid or alkaline that moist soils. Davis (1943) also investigated the effects of the treatment of the electrode prior to use. If the electrode was stored in an





alkaline solution, pH values in dry soils were higher than in moist soils. When storing the electrode in acid solutions or water, the pH values in dry soils were lower than in wet soils.

Schaller and Fischer (1981) observed lower pH values in slightly moistened soil samples compared to water-saturated samples
and concluded that the pH value decreases with increasing soil water tension. Adamchuk et al. (1999) developed an automated system for in situ pH measurements and tested it on soils in Indiana (USA). They observed a slight pH increase with increasing SMC. However, this variation was within the 95 % confidence interval of a standard soil pH measurement. Hence, they concluded that pH can be accurately measured in situ at SMC ranging between 15 to 25% for sandy soils and 20 to 30% for clayey soils. Oliviera et al. (2018) evaluated the influence of soil moisture on pH determination using antimony ISE on tropical
soils. They found an exponential relationship between pH and SMC and concluded that SMC influenced the electrode output of the pH ISE mainly when SMC was low. With increasing SMC, the pH value increased and finally stabilized at a SMC > 25 %. Furthermore, they observed a strongly increased dispersion of the pH measurements at low SMC. Another study on Oxisoles in São Paulo (Brazil) by de Souza Silva and Molin (2018) reported that SMC interferes with the readings of antimony pH ISEs in the manner that pH values proportionally increased with increasing SMC by about 0.9 units. Thus, they concluded that SMC
should be considered an issue when measuring pH in situ. However, they highlighted the need for additional studies on a range of different soil landscapes. Patil et al. (2019) investigated the pH response to SMC changes from 5 to 40 % in a Red and Bentonite soil in India. They found linearly increasing pH values of 1 and 1.7 units with increasing SMCs for Red and Bentonite soils, respectively. Finally, Zong et al. (2021) investigated the influence of SMC on antimony ISE pH readings in the laboratory. At 1 % SMC, the pH value was slightly higher than at 3% SMC, which gave the lowest readings. Then, the pH
strongly increased from 4.8 to 7.6 between 3 and 7 % SMC and reached a plateau from 7 % to 23 % SMC with only a minor pH increase of 0.5 units.

Thomas (1996) states, that the general increase of pH with SMC is much lower than one would expect, i.e. a 10-fold increase of SMC does not give a pH increase of one unit but often only about 0.4. He explains that in acid soils, the addition of water increases the dissociation of H+ from soil surfaces and increases the hydrolysis of Al species. This creates a buffer effect,
which maintains pH at a relatively stable value over a wide range of soil-water ratios. In soils with higher pH, hydrolysis of basic cations creates a similar buffer effect. Thomas (1996) concludes that these buffer effects are responsible for the observation, that water-soil ratios are not a highly important factor to consider when interpreting soil suspension pH values. However, this statement applies only for suspensions with a moisture content of 100 % or more.

The question can be raised, whether the deviation of in situ pH values from laboratory measurements in suspensions, in
particular at lower SMC, might be just artefacts of the potentiometric measurement principle caused by e.g., the selective bounding mechanisms of the membrane with $H^+$, influences on the salt bridge, or problems with the electrical signal amplifier in the instrument. However, this seems to be not the case because similar deviations were observed with pH sensors based on different measurement principles (Kahlert et al., 2004; Matthiesen, 2004; Merl et al., 2022). Merl et al. (2022) compared an optical pH sensor (pH optode) with a conventional glass pH electrode at different soil moisture levels down to 5 % and observed
a similar response of the pH readings in response to the SMC. While pH glass electrodes can suffer from failing connectivity



between reference and measurement electrode in very dry soils, pH optodes do not have this problem (Merl et al., 2022). They conclude that proton activity at low SMC, where the standard assumptions of aqueous solutions are no longer valid, are not fully understood and further investigations are needed. In absence of a sufficient theory and according algorithms for predicting the dependencies of soil pH at lower moisture levels, empirical studies play a major role in describing the phenomenon.

The main goal of the present paper is to study the effects of different soil textures and SMC on sensor-based pH value measurements using two ion-selective pH electrodes, i.e., an antimony and an epoxy-body electrode, on soils of a quaternary landscape of Northeast Germany. Specific objectives are: (i) to determine general trends of the pH response behaviour under varying soil textures and SMC, (ii) to identify SMC-related measurement discrepancies between the two pH electrodes, and (iii) to derive recommendations for a robust and precise sensor-based in situ soil acidity mapping under varying soil moisture

conditions.

## 2 Materials and methods

### 2.1 Study area and soil samples

Soil samples with varying soil texture were studied from arable land in a quaternary landscape of northeast Germany (Fig. 1A). This region was largely shaped by the Pleistocene glaciations and the Scandinavian inland ice sheet most of all by the

youngest Weichselian (115-12 ka) and the preceding Saalian glacial belt (150-130 ka; Krbetschek et al., 2008). Glacial, periglacial and interglacial processes created a mosaic of landforms and unconsolidated sediments, which tend to vary in physical and chemical properties over small distances. During the Holocene, soil formation is additionally controlled by land use. Predominant soil types are Cambisols, Luvisols, and Podzols on till plains and terminal moraines, Arenosols on glaciofluvial sands and aoelian sands as well as Gleysols, Histosols, and Fluvisols in groundwater-influenced valleys and

basins (Janetzko and Schmidt, 2014). Soil textures range from pure sand (class: Ss) to loamy clay (class: Tl) showing a dominance of sand and loam (classes: Sl, Su, St, Ls) according to the German soil classification system (KA5, Eckelmann et al., 2005). Soil pH values naturally range from acidic to alkaline as carbonates from glacial tills may occur in some places. Especially under agricultural use, soil pH is anthopogenically adjusted by lime fertilization and soils are predominantly poor in soil organic matter. Following the Koeppen-Geiger climate classification system, the study area is classified as temperate

oceanic with an increasing influence of continental circulations. The mean annual temperature is around 9 °C, with January being the coldest and July being the warmest month. With a total annual precipitation of 550 mm, the area is one of the driest regions in Germany.



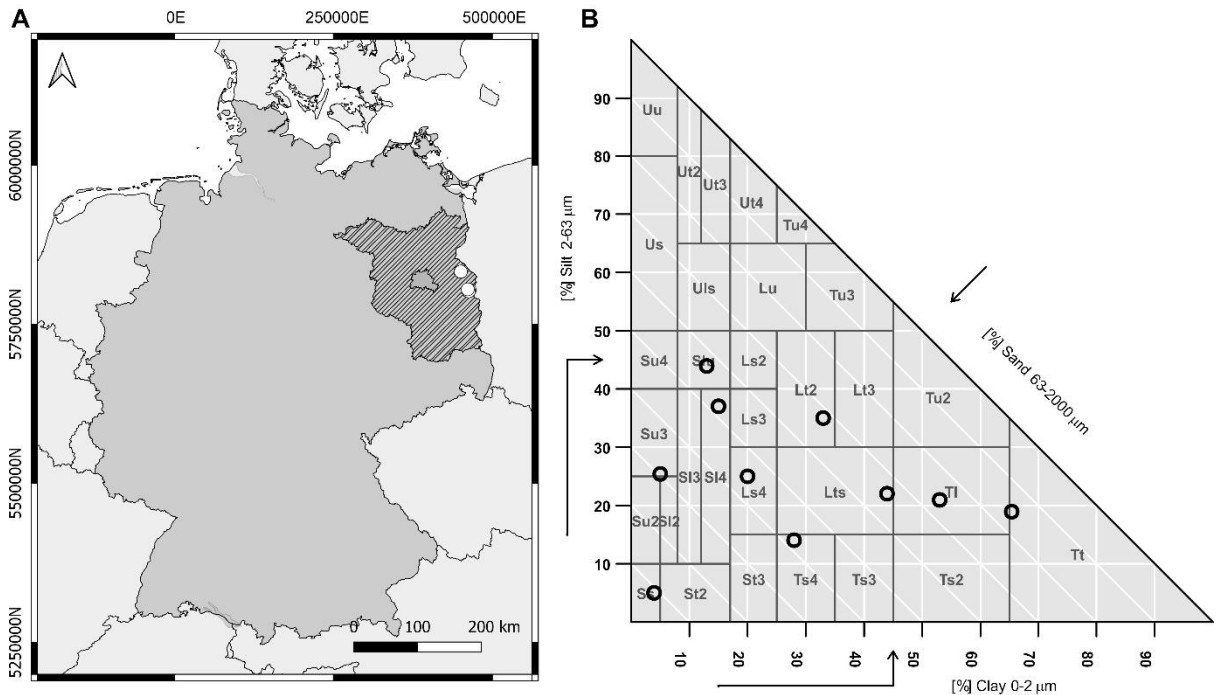

**Figure 1. A: Map of Germany with the location of the study sites (white dots) in the federal state of Brandenburg (shaded area), B: Classification of the 10 soil samples using the soil texture triangle of the German soil survey manual KA5 (Eckelmann et al., 2005).**

In order to determine the effect of different soil moisture contents and soil textures on sensor pH measurements, ten soil samples with a great range of values of the grain size fractions sand, silt and clay were selected using the soil texture triangle of the German soil survey manual KA5 (Eckelmann et al., 2005; Fig. 1B, Table 1). This includes the following 10 soil textures: sandy sand (Ss), strongly loamy sand (Sl4), medium silty sand (Su3), silty loamy sand (Slu), strongly sandy loam (Ls4), sandy clayey loam (Lts), slightly clayey loam (Lt2), strongly sandy clay (Ts4), loamy clay (Tl) and clayey clay (Tt). They cover 3 out of 4 soil texture main groups (sand, loam and clay) and 6 out of 11 soil texture groups (pure sand, silt sand, sand loam, normal loam, clay loam and loam clay; Eckelmann et al., 2005). Hence, sand, silt and clay contents range from 16 to 91 %, 5 to 44 % and 4 to 65 %, respectively.

## 2.2 Measurement setup and procedure

For baseline comparison, at first, the standard laboratory pH value of the soil samples was measured in 10 g of soil and 25 ml of 0.01 M $CaCl_2$ (soil:solution ratio = 1:2.5) with a glass electrode after 60 min (DIN ISO 10390). In order to measure the pH values of the soil samples under different soil textures and SMC, an individual laboratory-based measurement setup was



developed and applied on oven-dried (70 °C) and sieved (< 2mm) soil samples. It is schematically illustrated in Fig. 2. Two
pH sensors were used: a) an antimony electrode by Geoprobe (Geoprobe Systems, Salina, KS, USA) and b) a ruggedized glass
membrane electrode with a spear tip packaged in an epoxy body (hereafter called epoxy-body electrode) by PCE Instruments
(PCE Deutschland GmbH, Meschede, Germany).

The membrane of the antimony electrode consists of a thin layer of antimony trioxide ($Sb_2O_3$), formed by oxidation of the

surface of a cylindric piece of antimony at the tip of the electrode. It has an Ag/AgCl system as reference electrode. The
electrode potential is generated by the reaction of the $Sb_2O_3$ layer with the H+ ions in the sample solution. Antimony electrodes
can be used to reliably measure the pH in a range between 3 and 11 (Parks and Beard, 1933; Bates, 1961). The epoxy-body
electrode is made of a rugged epoxy body and has a glass membrane formed in a spear shape, which served as the measuring
tip. Thus, it is more resistant to mechanical stress compared to standard glass electrodes with glass bodies. It consists of an

Ag/AgCl reference electrode and covers a pH measuring range of 1 to 13.

For the analysis, 100 g of soil were weighed into a 125 ml sample cup that has a perforated lid at the underside. This allows
excess water to leave the soil into a sample cup underneath. A filter paper was placed on the perforated lid of the cylinder to
protect sample material from being washed out. The two pH ISE were fixed side by side in a laboratory stand and connected
to a self-developed ISE hardware and data acquisition software on a measuring computer. The setup is used for sensor

calibration and data logging, and was especially designed for measuring simultaneously with multiple ISEs. The ISE hardware
has a high impedance input amplifier for measuring the low output voltages of the antimony electrode.

Before each set of measurements, the pH electrodes were calibrated at the beginning of a measurement day using pH 7.01 and
pH 4.01 buffer solution (Hanna Instruments). The electrodes were immersed into the buffer solution for two minutes for the
pH readings to stabilize. The sensor pH values of the buffer solutions were simultaneously measured for both pH ISEs and a

calibration line was generated and displayed by the software. After each pH measurement, the sensor heads were cleaned with
distilled water to prevent from carry-over effects.

The very first pH measurements were conducted on dry soil samples with only the final droplet of distilled water from the
cleaning remaining on the sensor heads. Measurements were conducted for 30 seconds with five repetitions. The pH data were
automatically recorded seven to eight times per second and continuously displayed in a diagram. After completing the

measurements, the samples were oven-dried at 70°C.

After drying, 4 ml of distilled water was added to the same soil sample. In order to obtain a homogeneous distribution of the
water in the sample, the pH values were measured after a settling time of 30 minutes. After pH measurement, the sample cup
with the moist sample was weighted, oven-dried at 70°C and reweighted to determine its exact gravimetrical SMC.

In a next step, the previous amount of distilled water was increased by 4 ml, adding 8 ml to the dried sample and the pH values

were measured again after 30 minutes of equilibration. After weighting, drying and reweighting, the procedure began again.
The added water was continuously increased by 4 ml until the maximum capacity of capillary water was obtained in the sample
(near field capacity) and gravitational water began to flow out of the sample via the perforation at the bottom of the sample
cup. The same procedure was repeated for all ten soil texture samples.





It has to be emphasized that the soil samples are disturbed and capillary water is only held by the texture-related primary

structure of the soil. Consequently, the maximum capacity of capillary water deviates from field capacity under undisturbed

conditions additionally taking into account the secondary or aggregate structure of the soil. The maximum SMC near field

capacity was 16% for sandy soils (21 ml of water added) and 29% for clayey soils (40 ml of water added).

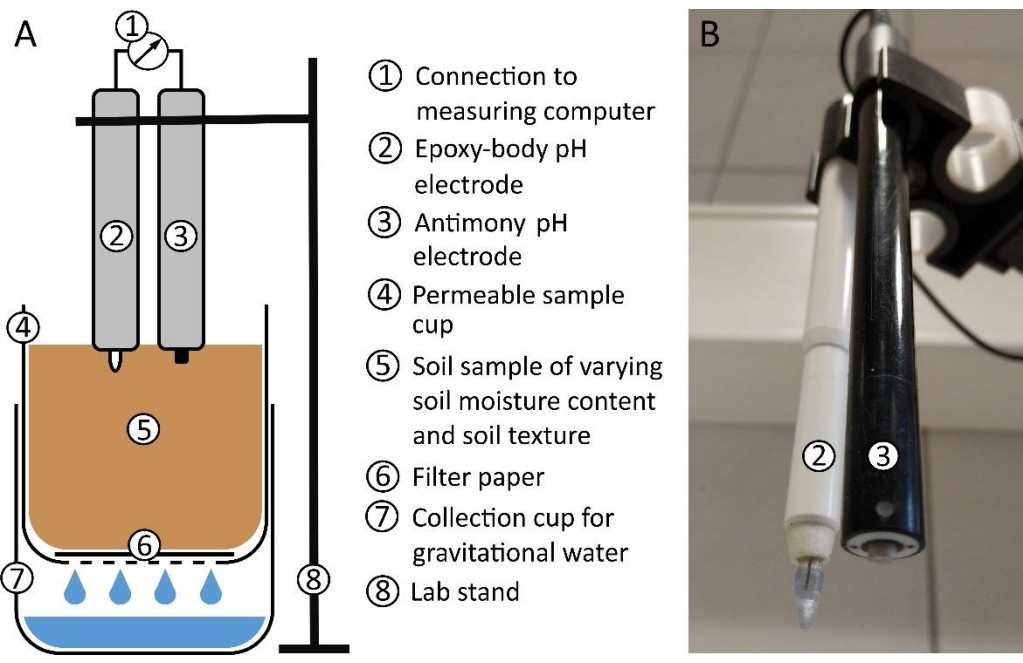

**Figure 2. Experimental setup (A) and photograph of the epoxy-body and antimony pH ISE.**

### 2.3 Data analysis

All data were processed and visualized in the free software environment for statistical computing and graphics R (version

4.1.0; R Core Team, 2018). Arithmetic means and standard deviations (σ) of the five repeated pH measurements were

calculated. The R functions lm and nls, implemented in the stats package (R Core Team, 2018), were used for fitting linear

and non-linear regression curves, respectively. The Spearman rank correlation coefficient (Spearman's Rho) was used to

quantify and compare linear and non-linear relationships.

To correlate the data with the soil texture, the mean particle diameter (MPD) of the fine earth fraction (< 2 mm) was calculated.

It was determined by the geometric mean of the particle diameter, which were obtained from transformation of the sand, silt

and clay contents (Shirazi et al., 1988; Shiozawa and Campell, 1991) following Eq. 1:

$$MPD = exp \sum_{i=1}^{n} m_i \ln(d_i) \qquad (1)$$



where MPD is given in [$10^{-3}$ m], mi is the mass fraction of the particle size class i [kg kg$^{-1}$], and d$_i$ is the associated geometric mean diameter [$10^{-3}$ m]. The MPD was established by Shirazi et al. (1988) and applied in the context of soil pH management by Rühlmann et al. (2021). The MPD has the advantage to subsume the three grain size fractions sand, silt and clay into one
single value for soil texture characterization. As the MPD of the ten soil textures used are skewed, it was converted into phi scale following Krumbein (1934, 1938; Eq. 2):

$$Phi = -log_2(MPD) \qquad (2)$$

where phi is the negative logarithm to the base 2 of the mean particle diameter [$10^{-3}$ m]. This allows putting more emphasis on the finer grain sizes (Donoghue, 2016).

To approximate the SMC at which the mean pH readings of the ISE begin to stabilize and the standard deviation of repeated measurements is minimized, the knee points of the regression curves were detected using the 'kneer' package in R, which implements the 'kneedle' algorithm by Satopää et al. (2011). A knee of a graph can be considered an operating point, where the perceived cost to alter a system parameter is no longer worth the expected performance benefit (Satopää et al., 2011). In the context of the present study, a knee can be defined as the point from which the pH change at continuously increasing SMC
is negligible. For knee point detection, the mathematical definition of curvature is used, where for a continuous function f exists a standard closed-form K$_f$ (x), the curvature of f at any point is a function of its first (f') and second (f'') derivative (Satopää et al., 2011):

$$K_f(x) = \frac{f''(x)}{(1+f'(x)^2)^{1.5}} \qquad (3)$$

## 3 Results and discussion

Figure 3 shows the linear correlation between pH values measured with the epoxy-body and the antimony pH electrode. It has a Spearman's correlation coefficient of 0.84 and a regression line that is slightly deviating from the 1:1 line by having an intercept of 1.1 and a slope of 0.8. This indicates a non-conformance in the pH measurements of the two ISEs, which is increasing towards lower and higher pH values, respectively. It is ± 0.1 pH units between pH 4.7 and 5.7 and rises to 0.5 pH units at pH 7.5. From the scatterplot, it can be seen that a large fraction of pH data of the same soil texture class are clustered
around similar pH values. Hence, the non-conformance in pH data from the epoxy-body and antimony ISE may in parts be texture-related. Furthermore, it can be observed from the pronounced vertical distribution of pH values parallel to the y axis especially for Ss, Sl4 and Tl, that the measurements of the antimony ISE are scattered over a wider pH range compared to the epoxy-body ISE.





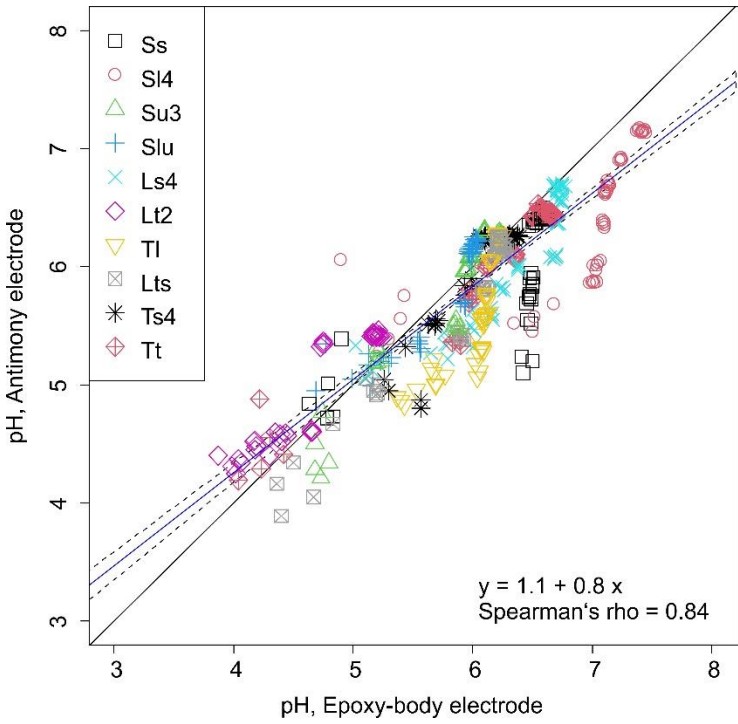

**Figure 3. Correlation between pH values measured with the epoxy-body and the antimony pH electrode in different soil textures and under different soil moisture regimes [solid black line: 1:1 line, solid blue line: linear regression model (dashed black lines: 95% confidence interval)].**

The arithmetic means and standard deviations of the five repeated sensor pH measurements at varying SMC and soil textures are shown in Table 1 and Figs. 4 and 5. The corresponding data table with all measured pH values can be found in the supplementary material (Tables S1, S2). In dry soils (0% gravimetric SMC), the epoxy-body electrode measured pH values between 4 (strongly acidic) and 5.5 (weakly acidic). Near field capacity, the pH increased to values between 5.2 (moderately acidic) and 7.4 (weakly alkaline). The measurement with the antimony electrode produced similar results measuring pH values

between 4.2 (strongly acidic) and 5.6 (weakly acidic) on dry soils and between pH 5.4 (weakly acidic) to 7.1 (weakly alkaline) near field capacity. The data show that with increasing SMC, pH values successively increased by approximately 1.5 pH units with both pH ISEs used. This pH increase is considerably higher compared to de Souza Silva and Molin (2018) who observed a pH increase of 0.9 units in Brazilian Oxisoles and within the upper range of 1 to 1.7 pH units as reported by Patil et al. (2019) from Red and Bentonite soils in India. Figure 6 illustrates that this maximum pH increase is soil texture dependent. It is lower

for sandy soils (higher mean particle diameter (MPD), lower phi-scaled MPD) and higher for clayey soils (lower MPD, higher phi-scaled MPD; Table 1). Thus, the influence of soil moisture variation on the sensor pH measurement is lower for sandy soils and higher for clayey soils.

The increase of the mean pH value with increasing SMC can be best described by an exponential model of the type:

$$pH = \alpha - \beta \cdot \gamma^{SMC} \qquad (3)$$



where α, β and γ are the regression coefficients of the exponential function describing the geometry of the graph. While α represents the maximum pH value at the soil's maximum amount of capillary water near field capacity, β is the difference between the starting point (SMC = 0) and the end point of the graph (near field capacity), i.e. the maximum pH increase. Finally, γ refers to the curvature of the graph. When γ is near 1, the graph has a nearly linear course and when γ approaches 0, the curve runs at an almost right angle exhibiting a strong concavity with a steep ascent.

For pure sand, the relation between pH and SMC is linear with γ being close to 1 irrespective of the pH ISE used. That means, the pH value increases constantly with increasing SMC over a wide range of moisture regimes. For the epoxy-body ISE and all other soil textures, γ ranges between 0.4 and 0.9 (Table 1), indicating a pronounced curvature of the fitted functions. In contrast, for the antimony ISE, a nearly linear curve, with γ close to 1 is additionally observed for the soil textures Su3, Slu and Lt2. For the other and especially the finer soil textures, the shape of the curve is also exponential (Table 1). As the mean

γ of the epoxy-body ISE is slightly lower compared to the antimony ISE, the exponential character of the pH curve obtained with the epoxy-body ISE is slightly more pronounced.

As can be seen in Figs. 4 and 5, the pH values measured during our experiment were always lower compared to the standard laboratory pH value measured with a glass electrode in 0.01 M $CaCl_2$ (soil:solution ratio = 1:2.5), irrespective of the pH ISE used and the soil texture and SMC the pH was measured in. In fact, with increasing SMC, the pH values approach this standard

pH asymptotically. Consequently, for both ISEs, the pH value near field capacity is very similar to the standard pH value of a soil:solution ratio of 1:2.5. However, since the graph of the epoxy-body ISE shows a stronger concavity (lower mean γ), it comes near the standard pH value earlier, i.e. at lower SMC (Figs. 4 & 5). This implies that, at high SMC near field capacity, subsequent calibration of the sensor pH values to the standard pH value becomes less necessary. This is particularly interesting if the sensor pH data is to be used for lime requirement estimates.

The observed exponential pH increase with respect to increasing SMC is consistent with findings of Oliviera et al. (2018) from tropical soils. However, besides for clayey soil textures, they found an exponential relationship also for sandy soils, which is in contrast to the present results. Patil et al. (2019) reported linearly increasing pH values for Indian Red and Bentonite soils. However, Bentonite soils are characterized by a high clay content, which could not be confirmed in the present study, where a linear correlation was only observed for sandy soils. Zong et al. (2021) also observed an exponential pH increase at increasing

SMC very similar to the observations for clayey textures in our study. However, in Zong et al. (2021), the soil texture was not reported.



**Table 1. Soil data and coefficients and Spearman'r rho of the exponential models as well as SMC at knee points.**

| ISE | Max. pH increase | Soil texture | Sand [%] | Silt [%] | Clay [%] | MPD [mm] | Phi | α | β | γ | Spearman's rho | SMC at knee point, Mean [%] | SMC at knee point, StdDev [%] |
|---|---|---|---|---|---|---|---|---|---|---|---|---|---|
| Epoxy-body | 1.17 | Ss | 90.7 | 5.2 | 4.1 | 0.216 | 2.21 | -13.2 | -17.2 | 1.0 | 0.92 | NA | 5.8 |
| | 0.71 | Su3 | 70.0 | 25.0 | 5.0 | 0.101 | 3.31 | 6.2 | 0.7 | 0.8 | 1 | 8.3 | 5.8 |
| | 1.08 | Sl4 | 47.7 | 36.8 | 15.6 | 0.048 | 4.38 | 6.1 | 1.2 | 0.9 | 0.9 | 8.4 | 5.9 |
| | 1.81 | Ls4 | 54.5 | 25.1 | 20.4 | 0.030 | 5.09 | 6.6 | 1.8 | 0.4 | 0.98 | 5.8 | 5.8 |
| | 1.59 | Slu | 43.3 | 43.8 | 13.0 | 0.028 | 5.14 | 6.9 | 1.7 | 0.9 | 0.95 | 8.5 | 8.5 |
| | 0.95 | Ts4 | 58.4 | 13.7 | 27.8 | 0.025 | 5.32 | 6.7 | 1.2 | 0.9 | 1 | 10.8 | 7.6 |
| | 2.18 | Lt2 | 31.7 | 35.1 | 33.3 | 0.008 | 7.00 | 7.3 | 2.0 | 0.8 | 1 | 7.9 | 11.0 |
| | 1.5 | Lts | 33.8 | 22.3 | 43.9 | 0.005 | 7.64 | 6.3 | 1.6 | 0.9 | 0.98 | 7.6 | 7.6 |
| | 1.67 | Tl | 26.3 | 20.7 | 52.9 | 0.003 | 8.48 | 6.3 | 1.8 | 0.8 | 0.88 | 10.9 | 7.6 |
| | 2.42 | Tt | 15.6 | 19.2 | 65.2 | 0.001 | 9.83 | 6.6 | 2.3 | 0.8 | 1 | 10.9 | 7.7 |
| Antimony | 1.08 | Ss | 90.7 | 5.2 | 4.1 | 0.216 | 2.21 | 1.2 | -3.1 | 1.0 | 0.92 | NA | 8.3 |
| | 1.33 | Su3 | 70.0 | 25.0 | 5.0 | 0.101 | 3.31 | 4.5 | -0.5 | 1.1 | 1 | NA | 10.9 |
| | 1.13 | Sl4 | 47.7 | 36.8 | 15.6 | 0.048 | 4.38 | 6.7 | 1.7 | 0.9 | 0.96 | 8.4 | 5.9 |
| | 1.47 | Ls4 | 54.5 | 25.1 | 20.4 | 0.030 | 5.09 | 6.7 | 1.9 | 0.9 | 0.88 | 8.4 | 8.4 |
| | 1.48 | Slu | 43.3 | 43.8 | 13.0 | 0.028 | 5.14 | -7.6 | -12.7 | 1.0 | 0.97 | NA | 10.9 |
| | 1.26 | Ts4 | 58.4 | 13.7 | 27.8 | 0.025 | 5.32 | 6.4 | 1.4 | 0.9 | 0.85 | 7.6 | 7.6 |
| | 1.54 | Lt2 | 31.7 | 35.1 | 33.3 | 0.008 | 7.00 | 1.7 | -3.8 | 1.0 | 0.99 | NA | 7.9 |
| | 1.85 | Lts | 33.8 | 22.3 | 43.9 | 0.005 | 7.64 | 6.5 | 2.1 | 0.9 | 0.93 | 7.6 | 7.6 |
| | 2.03 | Tl | 26.3 | 20.7 | 52.9 | 0.003 | 8.48 | 6.4 | 2.2 | 0.9 | 0.99 | 10.9 | 7.6 |
| | 1.98 | Tt | 15.6 | 19.2 | 65.2 | 0.001 | 9.83 | 6.6 | 2.1 | 0.9 | 0.82 | 10.9 | 7.7 |






**Figure 4. pH – SMC plots for different soil textures measured with the epoxy-body electrode and the antimony electrode. Left: Mean pH values, right: Standard deviations (n=5). Knee points are indicated by vertical lines.**






**Figure 5. pH – SMC plots for different soil textures measured with the epoxy-body electrode and the antimony electrode. Left: Mean pH values, right: Standard deviations (n=5). Knee points are indicated by vertical lines.**



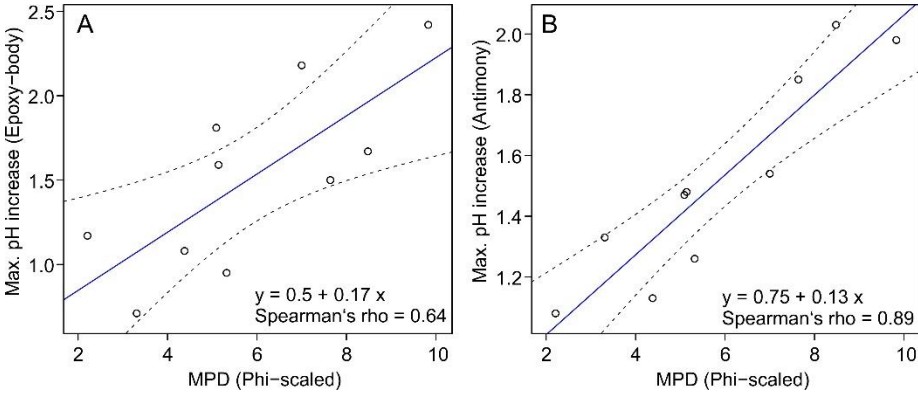

**Figure 6: Correlation between mean particle diameter (Phi-scaled) and maximum pH increase for the epoxy-body (A) and antimony electrode (B) [solid blue line: regression model, dashed black lines: 95% confidence interval].**

From the knee point detection of the exponential curves, it can be seen that stable pH reading are expected at a SMC between
6 % (Ls4) and 11 % (Tt) for the epoxy-body ISE and between 8 % (Ts4) and 11 % (Tt) for the antimony electrode (Table 1, Figs. 4, 5). Even though for Tl and Tt, the knee points were detected at the highest SMC, no clear relation to the soil texture was observed.

The threshold for obtaining constant pH measurements at 11 % SMC, as found in the present study, are much lower compared to previous work from Adamchuk et al. (1999), Kahlert et al. (2004) and Oliviera et al. (2018). They state that accurate pH measurement can be expected at a SMC of between 15 to 25% for sandy soils and between 20 to 30% for clayey soils (Adamchuk et al., 1999), between 10 and 40%, whereas at SMC < 10%, no reliable pH measurements could be carried out (Kahlert et al., 2004), and at SMC > 25% (Oliviera et al., 2018), respectively.

The standard deviations ($\sigma$) of the five repeated pH measurements in relation to SMC and soil texture shows high $\sigma$ at low SMC for both pH ISE and exponentially decreasing values towards higher SMC. Between 6 and 11% SMC, $\sigma$ is minimized and robust pH values can be measured with both pH ISEs and independent from soil texture (Table 1, Figs. 4, 5). Antimony ISE showed slightly higher $\sigma$ at dry soils and robust pH values at slightly higher SMC. These findings are in agreement with Keaton (1938) and Davis (1943) for glass membrane electrodes and with Oliviera et al. (2018) and Zong et al. (2021) for antimony electrodes, who also observed a strongly increased scattering of repeated pH measurements at low SMC. Zong et al. (2021) determined a threshold for minimized standard deviation at a SMC of 7%. A decreased pH dispersion at higher SMC was explained by Davis (1943) with the better soil contact at higher SMC. This is caused by the composition and operating principle of electro-chemical pH electrodes, requiring an ion-conductive connection between the reference and the measuring electrode, which is often not the case at dry conditions (Merl et al., 2022).



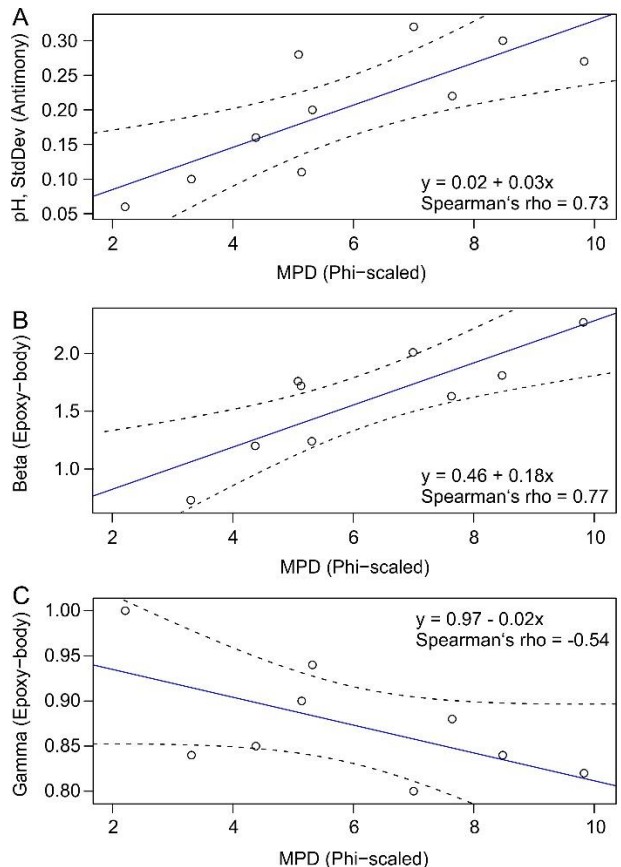

**Figure 7: Correlation between mean particle diameter (Phi-scaled) and standard deviation (A) as well as regression coefficients β (B), and γ (C) [solid blue line: regression model (shaded area: 95% confidence interval)].**

Figure 7 illustrates the dependence of soil texture represented by the phi-scaled MPD on the above-described SMC-pH relationship. For the antimony ISE and at low SMC, it can be seen that σ of the repeated pH measurements increases from 0.05 to 0.3 pH units with decreasing MPD and increasing phi (Fig. 7A). Thus, the measurement error is greater for clayey soils compared to sandy soils. Furthermore, also the shape of the exponential pH curves seems to be affected by soil texture. For the epoxy-body ISE, it was observed that the coefficient β, which represents the maximum pH increase of the exponential curve increases from 0.7 to 2.3 pH units when MPD decreases and phi increases (Fig. 7B). This confirms the above stated maximum pH increase calculated from the pH raw data (Table 1). Finally, γ, affecting the curvature of the model decreases with decreasing MPD and increasing phi (Fig. 7C). Hence, for sandy soil textures, the correlation between sensor pH and SMC is rather linear, whereas for clayey soils, the graph has a pronounced exponential geometry showing a strong concavity with a steep ascent at lower SMC and a plateau effect at higher SMC.

An explanation of the exponential relationship between pH and SMC was given by Thomas (1996). He states that an increase in SMC favours the dissociation of protons from the exchange sites of the soil as well as the hydrolysis of Al species at lower



pH values and of basic cations at higher pH values. These processes generate a buffer effect, which tend to stabilize the pH value at increasing SMC. Since, sandy soils have a smaller specific surface area and consequently lower cation exchange capacity, this buffer effect is less pronounced in sandy soils compared to clayey soils resulting in a rather linear SMC-pH relationship. In contrast, in clayey soils, the buffer effect results in a successive reduction of the pH increase with increasing SMC and in the described exponential behaviour.

**Conclusions**


In the present study, soil samples of 10 different soil textures from a quaternary landscape of Northeast Germany were exposed to different soil moisture contents (SMC) and the sensor pH values were measured with two different types of robust ion-selective electrodes. A change in soil moisture affected sensor pH readings, especially at low SMC and for soils with increasing amounts of clay. Whereas sandy soils rather show a linear relationship between pH and SMC or an exponential curve with low

curvature, in clayey soils, the concavity of the exponential model is more pronounced. The results show that reliable pH values are obtained for SMCs > 11%. The standard deviations ($\sigma$) of repeated measurements of both electrodes decreased with increasing soil moisture and showed a good precision at SMC > 11%. However, at low SMC, $\sigma$ was higher for clayey soils then for sandy soils. With increasing SMC, the pH values measured by ISEs approach the standard pH measured with a glass electrode in 0.01 M CaCl$_2$ (soil:solution ratio = 1:2.5) asymptotically. Thus, optimal measurement conditions can be considered

near field capacity, as a subsequent calibration of the sensor pH values to the standard pH value is negligible. In contrast, at low SMC, sensor data calibration is recommended. However, when the soil texture is known and the *in situ* soil moisture is measured, the regression curves can be used for sensor pH correction. Since, only 6 out of 11 soil texture groups were part of the present study, further analyses should be carried out on the 5 remaining, especially silty soil textures. Furthermore, more different soil landscapes should be involved.

**Author contribution**


Conceptualization: S.V.; Methodology: S.V., K.E.; Formal analysis: K.E., S.V.; Investigation: K.E., S.V.; Data curation: K.E., S.V.; Writing - original draft preparation: S.V., K.E., R.G.; Writing - review and editing: I.S., E.B., W.S., J.R., E.K.; visualization: S.V.; supervision: S.V., W.S.; Project administration: R.G., J.R., E.K.; Funding acquisition: R.G., J.R., E.K.

**Acknowledgments**

This work was partly funded by the agricultural European Innovation Partnership program (EIP-AGRI, Project: "pH-BB: precision liming in Brandenburg" (https://ph-bb.com), Project No.: 204016000014/80168341) as well as by the German Federal Ministry of Education and Research (BMBF, Project: "BonaRes (Modul A, I4S – Intelligence for Soil: Integrated





System for Site-specific Soil Fertility Management", Project No.: 031B1069A). Special gratitude goes to Mohamed Bourouah (Hahn Schickard Society for Applied Research) for providing the self-developed ISE hardware and software.

**Competing interest**

The authors declare that they have no conflict of interest.

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
