# Peer review of "The effect of soil moisture content and soil texture on fast in situ pH measurements with two types of robust ion-selective electrodes"

_EGUsphere, 2023_

## Author Response (AR1)

Author's response

**Review 1:**

Dear reviewer,

First of all I would like to thank you for reviewing our manuscript and your positive feedback.

Reviewer:

The used sensors are not well described.

Response:

Concerning the additional information about the sensors you demanded we will add two additional references in the text: "For more detailed information regarding glass and antimony pH ISE, the reader is referred to Fujimoto et al. (1980) and Schirrmann et al. (2011)."

Reviewer:

It could be nice to show a calibration cure of the sensors to see the response over the pH range. Also potential drift and need for recalibration could be evaluated.

Response:

Regarding the issue potential drift and recalibration we will add a new paragraph: "Ion-selective pH electrodes are generally considered to be reliable and accurate, but they can experience drift over time, which refers to a slow, gradual change in their response or calibration over time, leading to inaccurate pH measurements. Possible factors that can contribute to drift are: electrode aging, reference electrode issues, ion-selective membrane contamination, temperature changes, sample contamination or improper storage (Durst, 1978; Comer, 1991; Orellana et al., 2011). Regular calibration and maintenance are essential to minimize drift in ion-selective pH electrodes. Calibrating the electrode with standard buffer solutions, following proper storage and handling procedures, and replacing the electrode or its components when necessary can help maintain accuracy and reliability in pH measurements over time. The pH ISE should be calibrated at least at the beginning of each day or before each set of measurements. For in situ measurements, changing environmental conditions, such as major temperature fluctuations during the day can impact the electrode performance. In this case, it may be necessary to calibrate more often or perform a temperature compensation by integrating temperature measurements. Temperature and pH value are related, as the activity of ions in solution is temperature dependent. This relationship is described by the Nernst slope in the Nernst Equation (Barron et al., 2006)."

**Review 2:**

Dear reviewer,

Thank you very much for your review and the valuable remarks you have given therein. In the following, you find our answers to your specific comments:

Reviewer:

Line 206: The sample is not homogenized by shaking or similar. How is the homogeneity of the water content within the sample volume confirmed? It must be assumed that the tips of the electrodes "explore" a relatively small volume around them.

Resonse:

The homogeneous distribution of the added water within the soil sample was assured by using thoroughly homogenized soil samples and by allowing an equilibration time of 30 min. During that time, the matrix potential of the soil sucks the water into every direction of the soil volume.

Reviewer:

Line 226: Why is Spearman's Rho used and not the correlation coefficient r?

Response:

Lines 240 ff: The Spearman rank correlation coefficient (Spearman's Rho) was used in order to quantify and compare both linear and non-linear relationships. Furthermore, as it is based on the ranks of the data rather than actual values, Spearman's Rho can handle non-normally distributed data and is less sensitive to outliers compared to other correlation metrics.

Reviewer:

Line 333: At least in the case of the study by Oliveira et al 2018, I think the comparison of limit values (throughout the article) is inappropriate. In my opinion, in Oliveira's work, the humidity values are expressed based on the weight of dry soil (the authors do not explicitly indicate this but if this is not the case, the data referring to 100% humidity that appears in the graphs would not be possible). I think that in your article the moisture contents are expressed on the total weight of soil, that is, on a wet basis, although it should be specified in the Materials and Methods section. I am unaware of the studies by Adamchuk et al. (1999) and Kahlert et al. (2004) on what basis (dry or wet weight of soil) the moisture results are expressed but the authors should consider this.

Response:

Thank you very much for that valuable advice concerning the calculation basis of the soil moisture content (SMC). We have indeed calculated SMC on dry weight basis and specified it in the Materials and Methods section. I also checked the studies cited and added (when available) the basis of their SMC calculation in the text. Since all studies used for comparison of the threshold values used the SMC of dry weight basis, the comparison is indeed valid. Furthermore, the basis for calculation of SMC has not a strong impact at low SMC, because the deviation between SMC on dry and wet weight basis is negligible below a SMC of about 15%.

Reviewer:

Line 334: Without discarding Davis's explanation, perhaps the greater standard deviation for low moisture contents may be due to a lack of homogeneity in the distribution of water in the sample. It must be assumed that the surface of the electrode explores a certain space of the sample and with little water perhaps this distribution will be less effective.

Response

Thank you for your comment on the greater standard deviation at low moisture contents. We expect a lack of homogeneity in the distribution of water in the sample to be of minor importance. Even at low SMC, after the equilibration time of 30 min, the distribution of water in the sample around the pH electrode is expected to be as homogeneous as at high SMC as it is mostly driven by the matrix potential of the soil. However, we have integrated your argument in the text.